# SKINNY-Based RFID Lightweight Authentication Protocol

**DOI:** 10.3390/s20051366

**Published:** 2020-03-02

**Authors:** Liang Xiao, He Xu, Feng Zhu, Ruchuan Wang, Peng Li

**Affiliations:** 1School of Computer Science, Nanjing University of Posts and Telecommunications, Nanjing 210003, China; 1018041201@njupt.edu.cn (L.X.); xuhe@njupt.edu.cn (H.X.); zhufeng@njupt.edu.cn (F.Z.); wangrc@njupt.edu.cn (R.W.); 2Jiangsu High Technology Research Key Laboratory for Wireless Sensor Networks, Nanjing 210003, China

**Keywords:** RFID system, security protocol, SKINNY, mutual authentication, GNY logic

## Abstract

With the rapid development of the Internet of Things and the popularization of 5G communication technology, the security of resource-constrained IoT devices such as Radio Frequency Identification (RFID)-based applications have received extensive attention. In traditional RFID systems, the communication channel between the tag and the reader is vulnerable to various threats, including denial of service, spoofing, and desynchronization. Thus, the confidentiality and integrity of the transmitted data cannot be guaranteed. In order to solve these security problems, in this paper, we propose a new RFID authentication protocol based on a lightweight block cipher algorithm, SKINNY, (short for LRSAS). Security analysis shows that the LRSAS protocol guarantees mutual authentication and is resistant to various attacks, such as desynchronization attacks, replay attacks, and tracing attacks. Performance evaluations show that the proposed solution is suitable for low-cost tags while meeting security requirements. This protocol reaches a balance between security requirements and costs.

## 1. Introduction

The Internet of Things (IoT) is an object network that communicates with other objects through computers connected using the Internet, which can include any object with remote data collection, control, or communication capabilities, such as Automotive Cyber Physical Systems (ACPS), smart vehicles, home appliances, medical instruments, etc. In other words, the IoT involves many interrelated objects. Radio Frequency Identification (RFID) technology is one of the commonly used technologies in IoT and is widely used in various fields [1]. RFID technology integrates communication, storage, and computing components into accessible tags for wireless communication with readers over long distances. Each tag uniquely identifies its carrier while the carrier may be a product in a warehouse, a commodity in a retail store, an animal in a zoo, or a medical device in a hospital [2,3,4]. With the popularity of IoT technology, the scope of RFID applications has gradually expanded. Practical RFID systems are used in inventory and logistics management, object tracking, access control, automatic charging, anti-theft, localization, and intelligent transportation. According to market research by IDTechEx [5], the total RFID market in 2019 will reach $11.6 billion, and will increase to $13.4 billion in 2022. There exist various forms of passive tags and active tags, such as electronic tags, RFID cards, RFID readers, RFID keychains, and related software and services.

However, since the tag and the reader are wirelessly communicated in the RFID system, the technology suffers from security and privacy threats, i.e., an attacker can eavesdrop on the communication channel to achieve various attacks. The mutual authentication protocol is usually used to overcome the security attack between the reader and the tag. Since 2002, a lot of researches to secure RFID systems have been carried out, which are generally divided into four categories [6]: Mature protocol [7], simple protocol [8,9], lightweight protocol [10,11,12], and ultra-lightweight protocol [6,13,14]. Mature protocol refers to the protocols that require support for encryption algorithms in traditional cryptography, such as symmetric encryption, asymmetric encryption, and encrypted one-way functions; simple protocols apply to tags that support pseudo-random number generators and one-way hash functions; a lightweight protocol refers to a protocol whose tag can support pseudo-random number generator (PRNG) and simple functions such as cyclic redundancy code (CRC) check but does not support one-way hash function; ultra-lightweight protocol refers to a protocol that only involves simple bitwise logical operations such as XOR, AND, OR, etc. However, for RFID systems, the limitations of computing power and storage capacity, traditional cryptographic encryption protocols are difficult to apply to low-cost tags (5K–10K logic gates). Since ultra-lightweight protocols use only simple bit-wise operations, it is difficult to meet the security requirements. Furthermore, a large number of proposed ultra-lightweight protocols have been analyzed and attacked by other researchers [15], thus the use of relatively lightweight cryptographic algorithms to ensure the security certification of RFID systems is currently a research hotspot. 

Compared with traditional cryptographic algorithms, lightweight algorithms consume fewer resources during calculation and have a higher efficiency, which is very suitable for devices with limited computing capabilities such as RFID. Luo et al. [16] proposed a succinct and lightweight authentication protocol for low-cost RFID system. The authors claim that the protocol can resist various attacks, but Safkhani [17] proved that the protocol has desynchronization attack. Liu et al. [18] proposed an improved two-way authentication protocol for RFID systems. The author reduced the calculation and storage costs of tags by dividing the results obtained by the hash function into two parts, the left and right, to authenticate tags and readers. PRNG guarantees the dynamic update of keys and communication sub-messages, but the hash operation itself is computationally expensive, which is not suitable for low-cost tags. Gao et al. [19] proposed a lightweight RFID security authentication protocol based on the present encryption algorithm, but this protocol is not suitable for EPC C1 Gen2 compliant tags. Xu et al. [20] proposed a lightweight RFID two-way authentication protocol based on physical unclonable functions, using PUF and logical bit operations as security components. The protocol overcomes desynchronization attack by storing messages from the previous session. However, it has proved to be unable to resist a desynchronization attack and secret leak attack [21]. In addition, the stability of physical unclonable functions needs further research to improve. Zhang et al. [22] proposed a lightweight RFID group authentication protocol with strong track privacy protection. However, Gholami et al. [23] proved that the protocol could not resist a desynchronization attack and timeout problem.

In order to solve the above problems, this paper designs an RFID lightweight authentication protocol that meets the EPC standard based on the adjustable block cipher SKINNY algorithm. In this protocol, tags do not need to use hash functions and pseudo-random operations and rely on readers to complete complex pseudo-random operations, further reducing tag calculation costs. At the same time, the SKINNY encryption component guarantees the security of authentication and uses a dynamic update of the authentication sub-messages required for each session to resist tracking attacks. The security analysis proves that the protocol can resist most of the security threats currently existing in RFID systems.

The rest of this paper is composed as follows: In Section 2, the relevant symbol descriptions and a complete description of the protocol proposed in this paper are given. In Section 3, the security of the protocol is analyzed using GNY’s formal proof method and informal method. In Section 4, the four aspects of computing, communication, and storage, and security are compared with existing protocols. Finally, we conclude in Section 5.

## 2. LRSAS Protocol

### 2.1. Notations

To simplify the description, the symbols and operation instructions of the LRSAS protocol are shown in Table 1.

### 2.2. SKINNY Algorithm

The SKINNY algorithm is a lightweight block cipher proposed by Beierle et al., in 2016 [24], and its security structure belongs to the SPN cipher. SKINNY is a tweakable block cipher with multiple versions of block size and key size, which results in SKINNY being better adaptable to different application environments and having better performance in hardware implementation. Its block size n has 64-bit and 128-bit versions, and the key size t has n, 2n, and 3n versions. Since this paper studies the application in passive 96-bit-EPC-encoded RFID systems, the SKINNY encryption algorithm with a block size of 128 bits and a key size of n is used.

The SKINNY encryption algorithm includes three modules of initialization, the round function, and key scheduling. The encryption process of the three modules is briefly described below. The number of rounds of the SKINNY algorithm is shown in Table 2. In this paper, the block length is 128 bits, the key size is 128 bits, and the encryption round is 40 times.

Initialization. The 96-bit *FID* is divided into 16 8-bit sub-units, in which the high bits are zero-padded *FID* = *FID*_0_ ‖*FID*_1_ ‖⋯‖*FID*_14_ ‖*FID*_15_, in which *FID_i_* is an 8-bit plaintext subunit. This is represented by a row priority matrix, where *IS_i_ = FID_i_* for 0 ≤ *i* ≤ 15:$$IS=\left[\begin{array}{cccc} FID_{0} & FID_{1} & FID_{2} & FID_{3}\\ FID_{4} & FID_{5} & FID_{6} & FID_{7}\\ FID_{8} & FID_{9} & FID_{10} & FID_{11}\\ FID_{12} & FID_{13} & FID_{14} & FID_{15} \end{array}\right]$$

The initial key of 128 bits is represented by K, and K is divided into 8-bit sub-units thus that *K* = *K*_0_ ‖ *K*_1_ ‖…‖ *K*_14_ ‖ *K*_15_, in which *K_i_* is an 8-bit key subunit. The row priority matrix is used, where *TK_i_ = K_i_* for 0 ≤ *i* ≤ 15:$$TK=\left[\begin{array}{cccc} K_{0} & K_{1} & K_{2} & K_{3}\\ K_{4} & K_{5} & K_{6} & K_{7}\\ K_{8} & K_{9} & K_{10} & K_{11}\\ K_{12} & K_{13} & K_{14} & K_{15} \end{array}\right]$$

The Round Function. One encryption round of SKINNY is composed of five operations in the following order: SubCells, AddConstants, AddRoundTweakey, ShiftRows, and MixColumns. The number of rounds to perform depends on the block and key sizes.

Sub Cells(SC): The plaintext matrix *IS_i_* is nonlinearly transformed by the Sbox in units of single bytes. When the subunit is 8-bit, the Sbox is shown in Table 3 (in hexadecimal notation).

Add Constants(AC): The SC-transformed intermediate matrix is added to the round constant, and the round constant is generated by the linear shift register. The generation method can be referred to [24].

Add Round Tweakey(ART): The first 64-bit of the 128-bit intermediate matrix transformed by AC is xor with the first 64-bit of the round key, that is, *IS_i_ = IS_i_* ⊕ *TK_i_* for 0 ≤ *i* ≤ 7, where the round key passes through the key scheduling algorithm.

Shift Rows(SR): For the intermediate matrix of the ART transformation, the second, third, and fourth cell rows are rotated by 1, 2, and 3 positions to the right, respectively. In other words, a permutation P is applied: *P_T_*[*i*] = [0,1,2,3,7,4,5,6,10,11,8,9,13,14,15,12] for 0 ≤ *i* ≤ 15.

Mix Columns(MC): The SR-transformed intermediate matrix is right-multiplied by the matrix M.$$M=\left[\begin{array}{cccc} 1 & 0 & 1 & 1\\ 1 & 0 & 0 & 0\\ 0 & 1 & 1 & 0\\ 1 & 0 & 1 & 0 \end{array}\right]$$
The round function f(x) of the block cipher SKINNY-128-128 is shown in Figure 1.

Key Schedule. Suppose the key size is n, the key scheduling module is implemented by a permutation $P_{T}$, which is $P_{T}=[9,15,8,13,10,14,12,11,0,1,2,3,4,5,6,7]$. The content of 16 cells are replaced cell by cell according to the subscript rule indicated by $P_{T}$, thereby executing key updating.

### 2.3. LRSAS Protocol Description

In this protocol, passive RFID tags conforming to the 96-bit EPC code are used, which makes the tag limited by hardware and cost and cannot use traditional cryptographic encryption algorithms such as ECC and RSA. However, the lightweight block cipher SKINNY requires only 2391 logic gates under the premise of ensuring security, thus the SKINNY algorithm is very suitable for low-cost tags. The LRSAS protocol mainly includes four phases: Initialization phase, tag identification phase, mutual authentication phase, and update phase.

Initialization phase. There are three values inside each RFID tag: ID, FID, and K. ID and FID are 96-bit, K is 128-bit. FID and K are updated after each authentication. The back-end database will, respectively, store two sets of entries $\left\{ID,FID^{old},K^{old}\right\}$ and $\left\{ID,FID^{new},K^{new}\right\}$, which are the values communicated with the tag in the previous and current sessions, where FID is the pseudonym obtained by encrypting the ID using SKINNY.

Tag identification phase. The reader sends a request message, and the tag sends a response signal $FID^{new}$ to the reader after receiving the request signal. If the reader retrieves the data pair corresponding to $FID^{new}$ in the database, the authentication phase is entered; if the data pair corresponding to $FID^{old}$ is retrieved, the tag may be subjected to a desynchronization attack. In this case, the data pair $\left(FID^{old},K^{old}\right)$ is used for authentication.

Mutual authentication phase. The reader generates a random number r, calculates the message $M_{1}$ and $M_{2}$, and then sends $M_{1}\|M_{2}$ to the tag.
(1)$$M_{1}=FID⊕r$$
(2)$$M_{2}=E(FID⊕ID⊕r)$$

The tag calculates $r^{\prime }$ and ${M^{\prime }}_{2}$. If $M_{2}$ and ${M^{\prime }}_{2}$ are equal, the reader is authenticated. Otherwise, the authentication ends.
(3)$$r^{\prime }=M_{1}⊕FID$$
(4)$${M^{\prime }}_{2}=En(FID⊕ID⊕r^{\prime })$$

The tag calculates message ${M^{\prime }}_{3}$ and sends it to the reader.
(5)$${M^{\prime }}_{3}=En({M^{\prime }}_{2}⊕r^{\prime })$$

After receiving the message, the reader calculates $M_{3}$ according to its own $M_{2}$ and r. If $M_{3}$ and ${M^{\prime }}_{3}$ are equal, the tag is valid. Otherwise, the authentication ends.
(6)$$M_{3}=En(M_{2}⊕r)$$

Update phase. After the reader authenticates the tag, the session enters the updating phase. The reader sends OK information to the tag at the same time. Because the value of the last session tag is saved, the updating stage is divided into two situations. If the reader uses the $\left(FID^{old},K^{old}\right)$ pair to authenticate, the database will not update the pseudonym and shared key. If the reader uses the $\left(FID^{new},K^{new}\right)$ pair to authenticate, the database will update the pseudonym and the shared key in following way:(7)$$FID^{old}=FID^{new}$$
(8)$$K^{old}=K^{new}$$
(9)$$FID^{new}=M_{1}$$

The updating of the key $K^{new}$ is through the key schedule module in Section 2.2. After receiving the OK message, the tag updates its own pseudonym $FID^{new}=M_{1}$, and updates the key $K^{new}$ through the key schedule module, which is shown in Figure 2.

### 2.4. Formal Proof of the LRSAS Protocol

In this section, the GNY logic rules are used to prove the security and feasibility of the proposed LRSAS protocol. In this paper, the logical objects of GNY are tags and readers, which are represented by T and R, respectively. The key is represented by K. The formula variables are represented by X and Y. In order to simplify the structure of the article, the details of the GNY logic rules and symbolic representation can be found in [25].

(1) Protocol Initialization Assumption

Before using GNY logic to prove the proposed protocol, several necessary initial assumptions need to be given. Here is a list of specific assumptions:

P1: T∋(ID,FID,K)

P2: $R∋\left(ID,FID^{old},K^{old},FID^{new},K^{new},r\right)$

P3: T|≡ #(FID)

P4: R|≡ #(r)

P5: $T\overset{K,\;FID}{⟷}R$

(2) Establish an Idealized Protocol Model

M1: R→T: request

M2: T→R: FID

M3: R→T: FID⊕r||En(FID⊕ID⊕r)

M4: T→R: En(En(FID⊕ID⊕r′)⊕r′)

M5: R→T: confirmation 

The above description model can be converted into a model described using GNY logic language as follows:

M1: T ◁ request

M2: R ◁ FID

M3: T ◁ FID⊕r||En(FID⊕ID⊕r)

M4: R ◁ En(En(FID⊕ID⊕r′)⊕r′)

M5: T ◁ confirmation

(3) Protocol Target

The proof of the LRSAS protocol is to prove the freshness of the information sent by the other party when communicating with the reader and the reader. The target formula for the proof is as follows:
T |≡ R |~ #(FID⊕r, En(FID⊕ID⊕r))R |≡ T |~ #(En(En(FID⊕ID⊕r′)⊕r′))

(4) Protocol Reasoning of GNY Logic

According to GNY logic reasoning and initialization hypothesis, target 1 and target 2 are proved.

a. Proof target 1

According to the inference rule $\frac{A◁\left(X\right)}{A∋X}$ and the message M3, it can conclude:(10)T∋FID⊕r||En(FID⊕ID⊕r)

According to the inference rule $\frac{A◁\left(X,Y\right)}{A◁\left(X\right)}$ and the message M3, it can conclude:(11)T◁ FID⊕r
(12)T◁En(FID⊕ID⊕r)

According to the inference rule $\frac{A|\equiv B\overset{K}{⟷}A,A◁{\left\{X\right\}}_{K}}{A\left|\equiv B\right|\sim X}$, the assumption P5, and Formula (11), it can conclude:(13)T|≡R|~(FID⊕r)

According to the inference rule $\frac{A|\equiv B\overset{K}{⟷}A,A◁{\left\{X\right\}}_{K}}{A\left|\equiv B\right|\sim X}$, the assumption P5, and Formula (12), it can conclude:(14)T|≡R|~(En(FID⊕ID⊕r))

According to the inference rule $\frac{A|\equiv \#\left(X\right)}{A|\equiv \#\left(X,Y\right),A|\equiv \#\left(F\left(X\right)\right)}$, the assumption P3, it can conclude:(15)T|≡#(FID⊕r,En(FID⊕ID⊕r))

According to the Formulas (13)–(15), it can conclude: T |≡ R |~ #(FID⊕r, En(FID⊕ID⊕r))

b. Proof target 2

According to the inference rule $\frac{A◁\left(X\right)}{A∋X}$ and the message M4, it can conclude:(16)$$R∋En\left(En\left(FID⊕ID⊕r^{\prime }\right)⊕r^{\prime }\right)$$

According to the inference rule $\frac{A|\equiv B\overset{K}{⟷}A,A◁{\left\{X\right\}}_{K}}{A\left|\equiv B\right|\sim X}$, the assumption P5, and the message M3, it can conclude:(17)$$R\left|\equiv T\right|\sim En\left(En\left(FID⊕ID⊕r^{\prime }\right)⊕r^{\prime }\right)$$

According to the inference rule $\frac{A|\equiv \#\left(X\right)}{A|\equiv \#\left(X,Y\right),A|\equiv \#\left(F\left(X\right)\right)}$, the assumption P4, it can conclude:(18)$$R|\equiv \#\left(En\left(En\left(FID⊕ID⊕r^{\prime }\right)⊕r^{\prime }\right)\right)$$

According to the Formulas (17) and (18), it can conclude: $R\;\left|\equiv \;T\;\right|\sim \;\#\left(En\left(En\left(FID⊕ID⊕r^{\prime }\right)⊕r^{\prime }\right)\right)$


## 3. Informal Security Analysis

This section will analyze the security of LRSAS from seven security properties, including data confidentiality and integrity, replay attack, impersonation attack, tracking attack, desynchronization attack, denial of service attack, and forward security. The security of LRSAS is demonstrated by the following informal analysis.

Data confidentiality and integrity (DCI). In the authentication process, the (ID,K) of the tag and the r of the reader are transmitted in the form of ciphertext. Due to the security of the SKINNY packet encryption function and the pseudo-random number, the attacker cannot know the corresponding plaintext. In addition, the *FID* is that the tag’s pseudonym, which is updated after each successful session, thus the identity information of the tag is not leaked. In this protocol, the random number generation depends on readers with stronger computing capacity. In order to ensure that the random number received by the tag is the same as the random number generated by the reader, $M_{1}$ and $M_{2}$ contain r and ID. Encryption also guarantees the integrity. The reason is that any bit change of the random number r will result in different results of the ciphertext, leading to authentication failure.

Replay attack (RA). Since the tag and the reader communicate with each other through a wireless communication channel, an attacker can trick another subject by eavesdropping the transmitted sub-message, impersonating the tag or reader, and by replaying the previously received sub-message. It is assumed that the attacker records the information sent by the tag in advance. When the reader communicates with the tag again, the attacker pretends to be a legitimate tag and communicates with the reader through the recorded tag information. The values of FID and $M_{3}^{\prime }$. are related to the random number r of the reader. Since the random number of each authentication is different, each value of the tag response is different. Even if the illegal attacker intercepts the previous information, it cannot be used in the next time to forge the value. Therefore, the tag or reader will not accept the copied information.

Impersonation attack (IA). As discussed above, in the process of executing the LRSAS protocol, the tag and the reader need to be mutually authenticated, and the information used by the tag and the reader for mutual authentication is encrypted by the SKINNY algorithm, and the key is already stored in the initialization phase. In the main body, when an attacker wants to spoof another subject by forging one of the subjects, the correct ciphertext for verifying the identity information cannot be generated.

Track attack (TA). In each authentication phase, the tag does not transmit the plaintext of its ID or key, and the transmitted messages contain random numbers. In addition, the tag and database update the shared pseudonym FID and key K after each successful authentication. Second, no unbalanced operations, such as AND or OR operations, are used in the authentication protocol. Therefore, it is not feasible for an attacker to attack the current session by eavesdropping on historical information.

Desynchronization attack (DA). Since the tag and the background database update the pseudonym FID and the key K in each session, there is a problem that the shared data are inconsistent thus that the legitimate tag is subjected to the desynchronization attack, and thus cannot be authenticated in subsequent sessions. When the adversary tampers with the sub-messages $M_{1}$ and *M*_2_, the tag obtains an invalid random number $r^{\prime }$ through $M_{1}$, and then calculates $M_{2}^{\prime }$ through the wrong $r^{\prime }$. The tag authenticates the reader by comparing whether $M_{2}^{\prime }$ and $M_{2}$ are equal. Because the protocol guarantees the confidentiality and integrity of the message, the reader authentication fails in this session. The tag does not update information such as pseudonyms and keys and terminates the authentication. In addition, when the attacker interrupts $M_{3}$, the illegally generated $M_{3}$ will not pass the tag authentication, thus this protocol guarantees the synchronization of the information shared between the tag and the reader.

Denial of service attack (DoS). If the attacker blocks the final confirmation message sent by the reader, the adversary will cause a desynchronization attack. This problem can be overcome by storing the two versions of the (FID,K) values on the reader, storing the old version before the update, and storing the new version after the update. In addition, the tag can send an explicit ACK to confirm that the update phase was successful.

Forward security (FS). Since the pseudonym FID and shared key for authentication are updated after each session, and the pseudonym update needs to contain a random number. If the tag is cracked, the attacker cannot discover the historical confidential information. The previous communication of the tag and reader is still secure, which means forward security.

Compared with the security of the protocols proposed with the existing solutions, it can be clearly seen that compared with other protocols, the proposed protocol has the best security performance, as shown in Table 4.

From Table 4, the EMAP, SASI, and Gossamer protocols, which are ultra-lightweight protocols, are less secure than other lightweight and mature protocols in terms of secret disclosure attacks, denial of service attacks, and desynchronization attacks. Although the protocol based on the elliptic encryption curve achieves effective protection against common attacks, they need too much hardware resources due to the complexity of the mature encryption algorithm ECC calculation. The lightweight security protocols [16,19] reduce the consumption of hardware resources, but they cannot defend against synchronization attacks and tracking attacks. However, the LRSAS security protocol has reached a balance between security protection and resource consumption. Therefore, the LPSAS protocol has high availability and has a certain role in promoting the development of RFID security authentication protocols.

## 4. Performance Analysis

In the protocol proposed in this paper, the lightweight block cipher algorithm SKINNY was chosen as a security measure to ensure information confidentiality and integrity. Compared with the SIMON and PRESENT, which are common block ciphers, SKINNY not only has a lightweight key arrangement algorithm but also has the same efficiency as SIMON in execution [24]. This shows that SKINNY is very suitable for a low-cost RFID tag field. In addition, this protocol supports EPC coding for 96 bits. In the following, this paper compares and analyzes the protocol performance in terms of the communication overhead, storage overhead and computational overhead of the tag, as shown in Table 5.

Among them, h denotes a hash function operation, r denotes a random number generation operation, e denotes an ECC encryption/decryption operation, a denotes a connection operation, x denotes a logical bit operation, m denotes a MIXBITS operation in Gossamer, c denotes a Con encryption operation in SLAP, s denotes a SKINNY encryption operation, and p denotes a PRESENT encryption/decryption operation. The efficiency of encryption algorithm is x>s>p>m>c>h>e. In addition, L is the length of the pseudonym and key.

The protocol designed in this paper uses one of the SKINNY encryption algorithms and can support 96-bit EPC encoding. The calculation time of the round function used by the SKINNY encryption algorithm in the encryption phase is smaller than the Hash, ECC, and Present encryption calculation. Therefore, the calculation overhead is also applicable to low-cost RFID tags. In addition, the storage overhead of the tag is 3 L, which significantly reduces the storage capacity of the tag compared with other protocols, and lowers the complexity of the logic gate design of the storage structure. Furthermore, in the mutual authentication of the tag and the reader, the protocol has five information interactions, and the total amount of data received and transmitted is 6 L, which is relatively small, thereby ensuring the efficiency of information interaction.

Finally, in terms of the number of equivalent logic gates, different versions of SKINNY have different quantities of equivalent logic gates. This protocol uses SKINNY-128-128 version, the number of equivalent logic gates is 2391, less than 3K. Thus, it can be used in low-cost tags. In addition, the number of equivalent logic gates of other protocols also leads to being vulnerable to certain security attacks. See Table 4 for details.

## 5. Conclusions

This paper chooses a lightweight block cipher SKINNY, which has the advantages of low hardware power consumption and low computational complexity on the premise of ensuring secure encryption, thus it can be used in low-cost IoT terminal equipment. Based on the algorithm, this paper first designed a lightweight RFID security authentication protocol LRSAS, and then verified its security from seven security requirements, including data confidentiality and integrity, replay attack, impersonation attack, tracking attack, desynchronization attack, denial of service attack, and forward security, through GNY logic proof and informal security analysis. Finally, the performance analysis of LRSAS and other protocols was performed by comparing communication, storage, and computational overhead, which shows that the protocol can meet the security requirements and hardware overhead of the lightweight protocol.

## Figures and Tables

**Figure 1 sensors-20-01366-f001:**
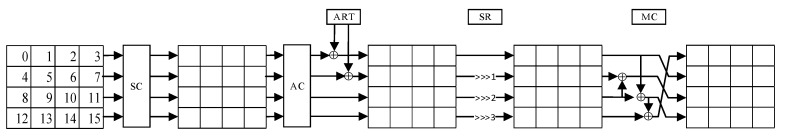
The SKINNY round function.

**Figure 2 sensors-20-01366-f002:**
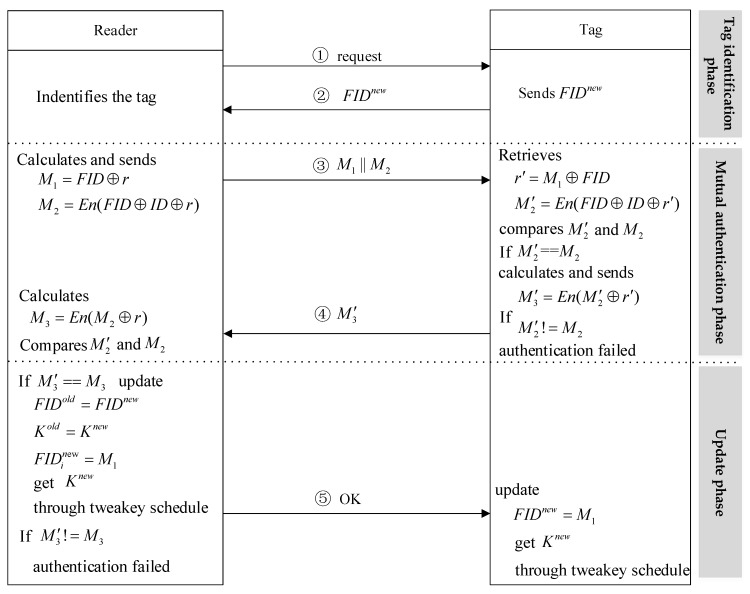
The authentication process of LRSAS.

**Table 1 sensors-20-01366-t001:** The description of notations.

Notations	Description
*R*	reader
*T*	tag
*ID*	unique identification of T
*FID*	pseudonym shared by T and R
*K*	key shared by T and R
*r*	random number generated by R
⊕	XOR operation
*En(X)*	SKINNY Encryption

**Table 2 sensors-20-01366-t002:** Number of rounds for SKINNY-n-t.

Block Size n / bit	Key Size t / bit	Round Times
64	64	32
	128	36
	192	40
128	128	40
	256	48
	384	56

**Table 3 sensors-20-01366-t003:** 8-bit Sbox $S_{8}$ used in SKINNY.

x	8bit (00~ff)															
$S_{8}[x]$	65	4c	6a	42	4b	63	43	6b	55	75	5a	7a	53	73	5b	7b
	35	8c	3a	81	89	33	80	3b	95	25	98	2a	90	23	99	2b
	e5	cc	e8	c1	c9	e0	c0	e9	d5	f5	d8	f8	d0	f0	d9	f9
	a5	1c	a8	12	1b	a0	13	a9	05	b5	0a	b8	03	b0	0b	b9
	32	88	3c	85	8d	34	84	3d	91	22	9c	2c	94	24	9d	2d
	62	4a	6c	45	4d	64	44	6d	52	72	5c	7c	54	74	5d	7d
	a1	1a	ac	15	1d	a4	14	ad	02	b1	0c	bc	04	b4	0d	bd
	e1	c8	ec	c5	cd	e4	c4	ed	d1	f1	dc	fc	d4	f4	dd	fd
	36	8e	38	82	8b	30	83	39	96	26	9a	28	93	20	9b	29
	66	4e	68	41	49	60	40	69	56	76	58	78	50	70	59	79
	a6	1e	aa	11	19	a3	10	ab	06	b6	80	ba	00	b3	09	bb
	e6	ce	ea	c2	cb	e3	c3	eb	d6	f6	da	fa	d3	f3	db	fb
	31	8a	3e	86	8f	37	87	3f	92	21	9e	2e	97	27	9f	2f
	61	48	6e	46	4f	67	47	6f	51	71	5e	7e	57	77	5f	7f
	a2	18	ae	16	1f	a7	17	af	01	b2	0e	be	07	b7	0f	bf
	e2	ca	ee	c6	cf	e7	c7	ef	d2	f2	de	fe	d7	f7	df	ff

**Table 4 sensors-20-01366-t004:** Security comparison.

Protocol	DCI	RA	IA	TA	DA	DoS	FS
EMAP [13]	×	√	√	√	×	×	×
SASI [6]	×	√	√	×	√	×	√
Gossamer [14]	√	√	√	√	×	×	√
ECC [7]	√	√	√	√	√	√	√
Present [19]	√	√	√	×	√	√	√
SLAP [16]	√	√	√	√	×	√	√
LRSAS	√	√	√	√	√	√	√

√: Satisfy, ×: Not satisfy.

**Table 5 sensors-20-01366-t005:** Performance comparison.

Overhead	EMAP [13]	SASI [6]	Gossamer [14]	ECC [7]	PRESENT [19]	SLAP [16]	LRSAS
communication	7 L	6 L	6 L	7 L	5 L	4 L	6 L
computational	22×	16×	32× + 3 m	H + r + 2 e + 2 s	4 p + a + r	9 c + 8× + a	4 s + × + a
storage	6 L	7 L	7 L	4 L	4 L	7 L	3 L

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
