# Peer review of "SKINNY-Based RFID Lightweight Authentication Protocol"

_sensors, 2020, doi:10.3390/s20051366_

Round 1
Reviewer 1 Report
The paper presents a new RFID lightweight authentication protocol based on a lightweight block cipher algorithm SKINNY and an RFID security authentication protocol.
The paper generally presents an interesting idea, but needs to be improved in the following aspects:
Further elaboration why such a protocol is needed should be added. A formal or semi-formal security proof needs to be added (e.g. ROR model, BAN logic, etc) Some verification of the protocol need to be added (e.g. AVISPA, ProVerif, Scyther, etc.) Simulation needs to be added for the performance analysis.
Author Response
Dear Professor Editor:
Thank you very much for the reviewers' kindly comments of Manuscript ID manuscript #sensors-686824 entitled " SKINNY-based RFID lightweight authentication protocol".
Based on the reviewers' valuable comments, we have made extensive modification on the original manuscript. Here, we attached revised manuscript in the formats of Doc and PDF document, for your approval. Here below is our description on revision according to the reviewers' comments.
Title: SKINNY-based RFID lightweight authentication protocol
Authors: Liang Xiao, He Xu, Feng Zhu, Ruchuan Wang, Peng Li *
Comments Reply to Reviewer #1
Further elaboration why such a protocol is needed should be added.
[Answer]
In the context of the Internet of Things, the realization of the Internet of Things requires the use of massive labels to identify object objects. However, attaching labels will increase the cost of items. Therefore, when mass objects need to be identified, the cost of labels will also be huge. Generally, in low-cost environments such as supply chains, low-cost passive tags are required.
In RFID systems, low-cost RFID tags (only 5k ~ 10k logic gates) are limited by hardware resources and costs. Many mature security algorithms with excessive hardware overhead cannot be applied to RFID systems, such as common public keys. Encryption algorithms RSA, ECC, etc. In addition, common hash functions also increase label costs. At present, RFID security protocols are roughly divided into four categories: mature protocols, simple protocols, lightweight protocols, and ultra-lightweight protocols. Among them, lightweight protocols and ultra-lightweight protocols can be applied to low-cost RFID systems, but a large number of studies have shown that ultra-lightweight protocols using only simple logical bit operations are difficult to meet the security requirements. However, the SKINNY algorithm for block ciphers only needs to occupy 2391 logic gates without losing security, and SKINNY is a This kind of adjustable block cipher can meet the requirements of RFID tags with 128-bit EPC coding. Therefore, SKNNIY algorithm is very suitable for the hardware conditions of RFID systems.
A formal or semi-formal security proof needs to be added (e.g. ROR model, BAN logic, etc). Some verification of the protocol need to be added (e.g. AVISPA, ProVerif, Scyther, etc.).
[Answer]
At present, the analysis methods of security protocols include intuitive manual analysis, actual attacks, and formal analysis methods. Among them, formal methods are widely used because of their strict mathematical theory, including modal logic, model checking, theorem proof, and process Algebra. BAN logic is one of the formal analysis methods of cryptographic protocols. Aiming at the shortcomings of BAN logic and other lacking an independent and clear semantic foundation, some improved and extended BAN-like logics are proposed, including GNY logic, AT logic, and VO Logic, etc. GNY logic provides a clearer modal theoretical semantics. This paper uses GNY logic to formalize the security analysis of the proposed protocol, which is described in detail in section 2.4 of the article. Since the formal analysis method GNY logic has been used to explain its security, we currently do not consider using other certification tools such as AVISPA, ProVerif, Scyther.
Simulation needs to be added for the performance analysis.
[Answer]
At present, we mainly analyze the theory and theoretically model the proposed lightweight protocol. We have not considered the part containing experimental simulation for the time being.
Should you have any questions, please contact us without hesitate. Thank you very much for your valuable comments.
Best regards,
Liang Xiao, He Xu, Zhu Feng, Ruchuan Wang, Peng Li

Reviewer 2 Report
This is an important contribution to the concept that Information Security has important implementation tradeoffs with an internet of things that connects powerful computer systems to small systems.
Author Response
Dear Professor Editor:
Thank you very much for the reviewers' kindly comments of Manuscript ID manuscript #sensors-686824 entitled " SKINNY-based RFID lightweight authentication protocol".
Based on the reviewers' valuable comments, we have made extensive modification on the original manuscript. Here, we attached revised manuscript in the formats of Doc and PDF document, for your approval. Here below is our description on revision according to the reviewers' comments.
Title: SKINNY-based RFID lightweight authentication protocol
Authors: Liang Xiao, He Xu, Feng Zhu, Ruchuan Wang, Peng Li *
Comments Reply to Reviewer #2
This is an important contribution to the concept that Information Security has important implementation tradeoffs with an internet of things that connects powerful computer systems to small systems.
[Answer]
Thanks to the reviewer’s valuable advice.
Should you have any questions, please contact us without hesitate. Thank you very much for your valuable comments.
Best regards,
Liang Xiao, He Xu, Zhu Feng, Ruchuan Wang, Peng Li

Reviewer 3 Report
Thie paper combines the SKINNY algorithm with the RFID security authentication protocol to offer a lightweight authentication protocol. The work is of significant importance to wireless communication and security advancements. Yet a number of improvements and clarifications are essential before considering this paper for publications.
+ The research gap is not clearly presented. Through a stronger literature review, the research gap must be presented. Currently many relevant literature specially for the year 2019 and after not included.
+What is the motivation and necessity of this paper with respect to the works of Wei Zhang et al., (2017), and Baolong Liu et al (2018)? please describe the differences and novelty of your paper.
+Why having table 3 is important? describe it in the paper, please.
+Why former work of the author on this topic is not cited and reviewed? It is essential that Xu et al., (2018) be reviewed and studied in the paper and the differences to be discussed.
+figure two must be given in an editable form instead of image places. Proper citation to the whole or part of it is required. Numbering the equations and descriptions of them in the text can be useful. Currently the operation not clear.
+ a detailed description on the given security comparison is necessary. Please elaborate on various protocols.
+native proofreading required.
Author Response
Dear Professor Editor:
Thank you very much for the reviewers' kindly comments of Manuscript ID manuscript #sensors-686824 entitled " SKINNY-based RFID lightweight authentication protocol".
Based on the reviewers' valuable comments, we have made extensive modification on the original manuscript. Here, we attached revised manuscript in the formats of Doc and PDF document, for your approval. Here below is our description on revision according to the reviewers' comments.
Title: SKINNY-based RFID lightweight authentication protocol
Authors: Liang Xiao, He Xu, Feng Zhu, Ruchuan Wang, Peng Li *
Comments Reply to Reviewer #3
The research gap is not clearly presented. Through a stronger literature review, the research gap must be presented. Currently many relevant literature specially for the year 2019 and after not included.
[Answer]
We made relevant amendments in the first section of the original text, discussed the current relevant lightweight RFID authentication protocols, and pointed out their problems and the advantages of this article, including the 2019 literature, The changes are as follows:
Compared with traditional cryptographic algorithms, lightweight algorithms consume less resources during calculation and have higher efficiency, which is very suitable for devices with limited computing capabilities such as RFID. Luo et al. [12] proposed a Succinct and Lightweight Authentication Protocol for low-cost RFID system. The authors claim that the protocol can resist various attacks, but Safkhani [19] proves that the protocol has desynchronization attacks. Masoumeh et al. [11] proposed a low-cost tag security authentication protocol based on the hash function. The author reduced the calculation cost by dividing the result obtained by the hash function into two parts for authentication and update, but the hash operation itself is expensive to calculate, etc The number of effective logic gates exceeds all logic gates owned by low-cost tags, which is not applicable to low-cost tags [13]. Gao et al. [20] proposed a lightweight RFID security authentication protocol based on the Present encryption algorithm, but this protocol is not suitable for EPC C1 Gen2 compliant tags.
In order to solve the above problems, this paper designs an RFID lightweight authentication protocol that meets EPC coding based on the adjustable block cipher SKINNY algorithm. In this protocol, tags do not need to use hash functions and pseudo-random operations, and rely on readers to complete complex pseudo-random operations, further reducing tag calculation costs. At the same time, the SKINNY encryption component guarantees the security of authentication, and uses a dynamic update of the authentication sub-messages required for each session to resist tracking attacks. The security analysis proves that the protocol can resist most of the security threats currently existing in RFID systems.
The rest of this paper is composed as follows: In Section 2, the relevant symbol descriptions and a complete description of the protocol proposed in this paper are given. In Section 3, the security of the protocol is analyzed using GNY's formal proof method and informal method. In Section 4, the four aspects of computing, communication and storage, and security are compared with existing protocols. Finally, we conclude in Section 5.
What is the motivation and necessity of this paper with respect to the works of Wei Zhang et al., (2017), and Baolong Liu et al (2018)? please describe the differences and novelty of your paper.
[Answer]
We have added a review of related papers in the first section of the article, which are described as follows:
Wei Zhang et al. (2017) proposed a lightweight RFID group authentication protocol with strong trajectory privacy protection. Its purpose is to improve the efficiency and security of tag group authentication. Its security components are mainly pseudo-random number generators and XOR operation. However, Gholami et al. (2019) point out that the protocol is not resistant to denial of service attacks, and there are also timeouts and forward security issues. Baolong Liu et al. (2018) proposed an improved two-way authentication of RFID system. The result of hash function is divided into left and right parts for authenticating tags and readers to reduce tag calculation and storage costs, using pseudo-random numbers. The generator guarantees the dynamic update of keys and communication sub-messages, thereby resisting retransmission attacks, tracking attacks, and clone attacks.
We propose an RFID security authentication protocol based on the block cipher SKINNY, in which the tag does not need to perform hash operations and pseudo-random operations, and uses a lightweight block cipher as its security component to ensure its data encryption security requirements. We rely on the reader to generate random numbers, which can further reduce the computational cost of tags. The difference from the protocol proposed by Zhang et al. Is that Zhang's protocol is aimed at security threats to label group authentication, while the proposed protocol is aimed at single-label authentication. The difference from Liu et al. Is that the tags in our protocol do not use expensive hash operations and PRNG operations.
Why having table 3 is important? describe it in the paper, please.
[Answer]
S-box is one of the core functions of the round function of block cipher SKINNY, which is used for non-linear permutation operation. The 8-bit S box can be described by 8 NOR operations and 8 XOR operations. In the SKINNY-128-128 version, an 8-bit S-box is used to perform a non-linear replacement operation on the input subunit data. If is the input 8-bit data bit (the lowest bit) and the output bit is , the S box changes as follows:
According to the above formula, the detailed data transformation of the 8-bit S-box transformation can be obtained as shown in Table 3 in the original text. Wherein the data is expressed in hexadecimal, the subunit in the correspondence function, and its range is from 00 to ff.
Table 3. 8-bit Sboxused in SKINNY
| 
 8bit(00~ff)  | 
||||||||||||||||
| 
 65  | 
 4c  | 
 6a  | 
 42  | 
 4b  | 
 63  | 
 43  | 
 6b  | 
 55  | 
 75  | 
 5a  | 
 7a  | 
 53  | 
 73  | 
 5b  | 
 7b  | 
|
| 
 35  | 
 8c  | 
 3a  | 
 81  | 
 89  | 
 33  | 
 80  | 
 3b  | 
 95  | 
 25  | 
 98  | 
 2a  | 
 90  | 
 23  | 
 99  | 
 2b  | 
|
| 
 e5  | 
 cc  | 
 e8  | 
 c1  | 
 c9  | 
 e0  | 
 c0  | 
 e9  | 
 d5  | 
 f5  | 
 d8  | 
 f8  | 
 d0  | 
 f0  | 
 d9  | 
 f9  | 
|
| 
 a5  | 
 1c  | 
 a8  | 
 12  | 
 1b  | 
 a0  | 
 13  | 
 a9  | 
 05  | 
 b5  | 
 0a  | 
 b8  | 
 03  | 
 b0  | 
 0b  | 
 b9  | 
|
| 
 32  | 
 88  | 
 3c  | 
 85  | 
 8d  | 
 34  | 
 84  | 
 3d  | 
 91  | 
 22  | 
 9c  | 
 2c  | 
 94  | 
 24  | 
 9d  | 
 2d  | 
|
| 
 62  | 
 4a  | 
 6c  | 
 45  | 
 4d  | 
 64  | 
 44  | 
 6d  | 
 52  | 
 72  | 
 5c  | 
 7c  | 
 54  | 
 74  | 
 5d  | 
 7d  | 
|
| 
 a1  | 
 1a  | 
 ac  | 
 15  | 
 1d  | 
 a4  | 
 14  | 
 ad  | 
 02  | 
 b1  | 
 0c  | 
 bc  | 
 04  | 
 b4  | 
 0d  | 
 bd  | 
|
| 
 e1  | 
 c8  | 
 ec  | 
 c5  | 
 cd  | 
 e4  | 
 c4  | 
 ed  | 
 d1  | 
 f1  | 
 dc  | 
 fc  | 
 d4  | 
 f4  | 
 dd  | 
 fd  | 
|
| 
 36  | 
 8e  | 
 38  | 
 82  | 
 8b  | 
 30  | 
 83  | 
 39  | 
 96  | 
 26  | 
 9a  | 
 28  | 
 93  | 
 20  | 
 9b  | 
 29  | 
|
| 
 66  | 
 4e  | 
 68  | 
 41  | 
 49  | 
 60  | 
 40  | 
 69  | 
 56  | 
 76  | 
 58  | 
 78  | 
 50  | 
 70  | 
 59  | 
 79  | 
|
| 
 a6  | 
 1e  | 
 aa  | 
 11  | 
 19  | 
 a3  | 
 10  | 
 ab  | 
 06  | 
 b6  | 
 80  | 
 ba  | 
 00  | 
 b3  | 
 09  | 
 bb  | 
|
| 
 e6  | 
 ce  | 
 ea  | 
 c2  | 
 cb  | 
 e3  | 
 c3  | 
 eb  | 
 d6  | 
 f6  | 
 da  | 
 fa  | 
 d3  | 
 f3  | 
 db  | 
 fb  | 
|
| 
 31  | 
 8a  | 
 3e  | 
 86  | 
 8f  | 
 37  | 
 87  | 
 3f  | 
 92  | 
 21  | 
 9e  | 
 2e  | 
 97  | 
 27  | 
 9f  | 
 2f  | 
|
| 
 61  | 
 48  | 
 6e  | 
 46  | 
 4f  | 
 67  | 
 47  | 
 6f  | 
 51  | 
 71  | 
 5e  | 
 7e  | 
 57  | 
 77  | 
 5f  | 
 7f  | 
|
| 
 a2  | 
 18  | 
 ae  | 
 16  | 
 1f  | 
 a7  | 
 17  | 
 af  | 
 01  | 
 b2  | 
 0e  | 
 be  | 
 07  | 
 b7  | 
 0f  | 
 bf  | 
|
| 
 e2  | 
 ca  | 
 ee  | 
 c6  | 
 cf  | 
 e7  | 
 c7  | 
 ef  | 
 d2  | 
 f2  | 
 de  | 
 fe  | 
 d7  | 
 f7  | 
 df  | 
 ff  | 
|
Why former work of the author on this topic is not cited and reviewed? It is essential that Xu et al., (2018) be reviewed and studied in the paper and the differences to be discussed.
[Answer]
We have added a citation and review of the recent "A Lightweight RFID Mutual Authentication Protocol Based on Physical Unclonable Function" paper published by Xu et al. This thesis is based on a new type of hardware security primitive as the authentication security component, which is different from the block cipher SKINNY adopted as the authentication security component. In addition, PUF-based security protocols are difficult to quantify and are susceptible to attack methods such as model building attacks and power analysis attacks, and the stability of PUF due to environmental changes such as temperature and noise requires further research.
figure two must be given in an editable form instead of image places.
[Answer]
We have revised the page of Fig. 2. Thanks for the reviewer’s advice.
Proper citation to the whole or part of it is required. Numbering the equations and descriptions of them in the text can be useful. Currently the operation not clear.
[Answer]
We have revised the citation.We have revised the description of the equations in Section 2 and numbered the equations to make them look clearer.Thanks for the reviewer’s advice.
a detailed description on the given security comparison is necessary. Please elaborate on various protocols.
[Answer]
We give formal and informal security analysis of the proposed protocol in Sections 2.4 and 3 of the paper. We demonstrated the security of the proposed protocol from seven aspects: data confidentiality and integrity, retransmission attacks, impersonation attacks, tracking attacks, desynchronization attacks, denial of service attacks, and forward security. Compare security.
From Table 4, the EMAP, SASI and Gossamer protocols, which are ultra-lightweight protocols, are less secure than other lightweight and mature protocols. In terms of secret disclosure attacks, denial of service attacks, and desynchronization attacks The security is not perfect; although the protocol based on the elliptic encryption curve achieves effective protection against common attacks, there are limitations in hardware resources due to the complexity of the mature encryption algorithm ECC calculation; the lightweight security protocol [12] [20] The elliptic curve encryption algorithm reduces the consumption of hardware resources, but it cannot defend against synchronization attacks and tracking attacks. However, the LRSAS security protocol has reached a balance between security protection and resource consumption. Therefore, the LPSAS protocol has high availability and has a certain role in promoting the development of RFID security authentication protocols.
native proofreading required.
[Answer]
We have checked and revised some English errors and improves the form in the light of the reviewer’s advice.
Should you have any questions, please contact us without hesitate. Thank you very much for your valuable comments.
Best regards,
Liang Xiao, He Xu, Zhu Feng, Ruchuan Wang, Peng Li

Round 2
Reviewer 1 Report
The paper presents a new RFID lightweight authentication protocol based on a lightweight block cipher algorithm SKINNY and an RFID security authentication protocol.
In the revised version of the paper, the authors have adequately resolved the issues raised by the reviewers.
Author Response
Thank you very much for the reviewers' kindly comments of Manuscript ID manuscript #sensors-686824 entitled " SKINNY-based RFID lightweight authentication protocol".
Based on the reviewers' valuable comments, we have made extensive modification on the original manuscript. Here, we attached revised manuscript in the formats of Doc and PDF document, for your approval. Here below is our description on revision according to the reviewers' comments.
Title: SKINNY-based RFID lightweight authentication protocol
Authors: Liang Xiao, He Xu, Feng Zhu, Ruchuan Wang, Peng Li *
Comments Reply to Reviewer #1
In the revised version of the paper, the authors have adequately resolved the issues raised by the reviewers.
[Answer]
Thanks to the reviewer’s valuable advice.
Should you have any questions, please contact us without hesitate. Thank you very much for your valuable comments.

Reviewer 3 Report
A number of revisions have been applied. Yet the improvements are not adequate.
Mostly weak and wrong citations are used, e.g., the reference and citation [1] is not correct. This goes throughout the paper. please use accurate citations. Many claims are not cited.
In addition to the former comment, the order of citation and referencing are also not correct.
Figure two must be either presented as a table or goes through a major improvement.
I have a serious concern about reference 21 which is not cited in the paper properly. I also cannot find proper citation to the references 21-25.
Author Response
The Second Round Comments Reply
Dear Professor Editor:
Thank you very much for the reviewers' kindly comments of Manuscript ID manuscript #sensors-686824 entitled " SKINNY-based RFID lightweight authentication protocol".
Based on the reviewers' valuable comments, we have made extensive modification on the original manuscript. Here, we attached revised manuscript in the formats of Doc and PDF document, for your approval. Here below is our description on revision according to the reviewers' comments.
Title: SKINNY-based RFID lightweight authentication protocol
Authors: Liang Xiao, He Xu, Feng Zhu, Ruchuan Wang, Peng Li *
Comments Reply to Reviewer #3
- Mostly weak and wrong citations are used, e.g., the reference and citation [1] is not correct. This goes throughout the paper. please use accurate citations. Many claims are not cited.

[Answer]
The reference [1] is about backscatter communication, where RFID is one of the backscatter devices. This article describes some typical applications, including the use of RFID tags for all items in a store. Thus, we cited this article in the first section. Furthermore, we have checked and revised the citation and reference. Thanks for the reviewer’s advice.
- In addition to the former comment, the order of citation and referencing are also not correct.

[Answer]
We have reviewed and revised the order of citation and referencing. Thanks for the reviewer’s advice.
- Figure two must be either presented as a table or goes through a major improvement.

[Answer]
We have modified the layout of Figure 2 to make the authentication process clearer. The modified content is as Figure 2.
- I have a serious concern about reference 21 which is not cited in the paper properly. I also cannot find proper citation to the references 21-25.

[Answer]
Because the order of references has been adjusted, the current serial number may not be equal to the previous one. We have revised the first section of the article in the light of the reviewer’s advice, supplementing the literature review with references 21-24 as follows:
Xu et al. [21] proposed a lightweight RFID two-way authentication protocol based on physical unclonable functions, using PUF and logical bit operations as security components. The protocol overcomes desynchronization attacks by storing messages from the previous session. However, it has proved to be unable to resist desynchronization attacks and secret leak attacks [22]. In addition, the stability of physical unclonable functions needs further research to improve. Zhang et al. [23] proposed a lightweight RFID group authentication protocol with strong track privacy protection. However, Gholami et al. [24] proved that the protocol cannot resist desynchronization attacks and timeout problems.
Should you have any questions, please contact us without hesitate. Thank you very much for your valuable comments.
Best regards,
Liang Xiao, He Xu, Feng Zhu, Ruchuan Wang, Peng Li

Round 3
Reviewer 3 Report
The revised version can be considered for publication.
Please consider these minor corrections:
+ References style is not the MDPI standard.
+ Figure two must be in a frame
+Author Contributions section is not the MDPI standard.
+English proofreading is essential
+some parts are bold: please correct.
Author Response
The Third Round Comments Reply
Dear Professor Editor:
Thank you very much for the reviewers' kindly comments of Manuscript ID manuscript #sensors-686824 entitled " SKINNY-based RFID lightweight authentication protocol".
Based on the reviewers' valuable comments, we have made extensive modification on the original manuscript. Here, we attached revised manuscript in the formats of Doc and PDF document, for your approval. Here below is our description on revision according to the reviewers' comments.
Title: SKINNY-based RFID lightweight authentication protocol
Authors: Liang Xiao, He Xu, Feng Zhu, Ruchuan Wang, Peng Li *
Comments Reply to Reviewer #3
- References style is not the MDPI standard.

[Answer]
We have revised the references style following the MDPI standard. Thanks for the reviewer’s advice.
- Author Contributions section is not the MDPI standard.

[Answer]
We have revised the section of author contributions following the MDPI standard. Thanks for the reviewer’s advice. The modified content is as follows:
Author Contributions: Methodology, Liang Xiao and Peng Li; validation, Liang Xiao and He Xu; formal analysis, He Xu; writing—original draft preparation, Liang Xiao; writing—review and editing, Feng Zhu; funding acquisition, Ruchuan Wang. All authors have read and agreed to the published version of the manuscript.
- Figure two must be in a frame

[Answer]
We have modified Figure 2 in the light of the reviewer’s advice, which make the process clearer. As shown in the Figure 2. The authentication process of LRSAS
- English proofreading is essential

[Answer]
We have checked and revised some English errors and improves the form in the light of the reviewer’s advice.
- some parts are bold: please correct.

[Answer]
We have revised some bold text to remove the bold effect. Thanks for the reviewer’s advice.
